# Behavioural variability and repeatability in adult zebrafish (*Danio rerio*) using the novel tank dive test

Andréa L. Johnson[1], Peter L. Hurd[1,2], Kimberley J. Mathot[3], Trevor J. Hamilton[1,4,5]*

1 Neuroscience and Mental Health Institute, University of Alberta, Edmonton, Alberta, Canada,
2 Department of Psychology, University of Alberta, Edmonton, Alberta, Canada, 3 Department of Biological Sciences, University of Alberta, Edmonton, Alberta, Canada, 4 Department of Psychology, MacEwan University, Edmonton, Alberta, Canada, 5 Allen Discovery Center for Neurobiology in Changing Environments, University of California San Diego, San Diego, California, United States of America

* trevorjameshamilton@gmail.com

## Abstract

Zebrafish (*Danio rerio*) are widely used in behavioural neuroscience as a model for studying anxiety-like and stress-related behaviours. However, substantial variability exists within and among individuals, influenced by factors such as sex, age, and environmental conditions, making the interpretation of anxiety-related behaviours challenging. Here we characterized longitudinal patterns of stability and variability in anxiety-like behaviours across individual adult zebrafish and assessed whether distinct behavioural profiles emerged over time. Using the novel tank dive test, we tracked anxiety-related behaviours in zebrafish across multiple time points over a 21-week period (90, 120, and 150 days post-fertilization). Behavioural metrics, including time spent in tank zones, swimming velocity, and immobility, were analyzed for age- and sex-related effects, repeatability, and group variation. Results indicated significant changes in anxiety-like behaviours with age, with fish spending more time in the upper zone and displaying increased swimming velocity over time. While no significant sex differences were observed in zone preference, males exhibited greater within-individual variation in time spent in the lower zone, while females demonstrated higher among-individual variation and repeatability over time. Furthermore, zebrafish were classified into high, medium, and low-anxiety groups based on cumulative behavioural scores, revealing stable individual differences in anxiety-like behaviours. These findings highlight the importance of considering age, sex and both intra- and inter-individual variation when interpreting zebrafish behaviour and provide a foundation for future research exploring selective breeding, anxiety level interactions, and pharmacological modulation of anxiety-related phenotypes.

**Data availability statement:** All relevant data are within the paper and its Supporting information files.

**Funding:** This research received funding from the Natural Sciences and Engineering Research Council of Canada (NSERC) Discovery Development grant to T.J.H. (05426). This research was supported by the Allen Discovery Center program (T.J.H.), a Paul G. Allen Frontiers Group advised program of Allen Family Philanthropies. The funders had no role in study design, data collection and analysis, decision to publish, or preparation of the manuscript.

**Competing interests:** The authors have declared that no competing interests exist.

## 1. Introduction

Zebrafish (*Danio rerio*) are a widely recognized model in the medical sciences including behavioural neuroscience, and frequently used for investigating stress-related and anxiety-like behaviours. Their small size, ease of care, and significant genetic and physiological homology to humans make them an ideal model for investigating the neurobiology of anxiety and screening anxiolytic compounds [1–9]. In behavioural research, their stereotypic behaviours make them an ideal model for investigating pathologies and developing new therapies for neurological disorders [10,11]. Their high sensitivity to anxiety-evoking environmental stressors and anxiolytic manipulations [12] allows investigation of robust stress and drug-evoked phenotypic behaviours [13]. Furthermore, zebrafish exhibit an evolutionarily conserved stress response mediated by the hypothalamus-pituitary-interrenal (HPI) axis, which is homologous to the human hypothalamus-pituitary-adrenal (HPA) axis [12].

Zebrafish behavioural assays, including the novel tank dive test, open field test, and light-dark preference test are well-established to reliably measure anxiety-related behaviours [1,4,5,14–19]. These tests assess exploratory and locomotor behaviours, such as time spent in different zones of a testing arena, swimming velocity, and freezing bouts, which serve as proxies for anxiety levels [20]. However, despite the utility of these measures, behavioural variability within and among individual zebrafish remains a significant challenge for interpreting results. Replicating zebrafish behavioural studies can be challenging due to variability introduced by experimental setups, environmental conditions, and biological factors such as age, sex, strain, and environmental history, all of which influence anxiety-related behaviours [4,21–25]. For example, older zebrafish often exhibit reduced baseline locomotion compared to younger fish, and both age and sex differences have been shown to affect anxiety-like and locomotor behaviours [25,26]. However, sex effects in zebrafish behaviour are not uniform across strains or contexts. For example, Mustafa et al. [27] found that strain identity strongly modulated sex differences in boldness and anxiety-like behaviour. Additionally, laboratory strains generally display less pronounced anxiety-like behaviours compared to wild-type strains, while mutant strains often display heightened anxiety-like behaviours [4,23,24]. This variability is widely observed in natural populations and plays an essential role in a species' overall response to a change in environment [28]. However, such variation, both at the group level (e.g., strain, age, or sex differences) and at the individual level, can contribute to significant variability in experimental outcomes, making it essential to account for these differences when interpreting behavioural results or assessing repeatability in traits [28].

Recently, researchers have emphasized the importance of understanding both intra- and inter-individual variation in zebrafish behaviour [22]. While group-level averages provide valuable insights, they can obscure individual behavioural patterns and consistencies that are important for accurately interpreting experimental results and identifying meaningful behavioural trends [22,28]. Longitudinal studies have shown that anxiety-like behaviours in zebrafish can exhibit significant repeatability over time, with differences observed across sexes and experimental contexts [29]. For example,

male zebrafish often demonstrate higher repeatability in anxiety-related measures compared to females, suggesting potential sex-specific stability in behavioural traits [29,30]. However, these studies tested fish at 5 months and older, had relatively small sample sizes (24 males and 24 females), and used only one behavioural test. Another study examined individual stress-related behaviour in 21 month old zebrafish over a 5 week duration between wild-caught zebrafish and tenth generation zebrafish selectively bred for high and low stress-coping style [31]. On average, stress-related behaviour was consistent and repeatable across time in all groups. Notably, the selectively bred strains showed higher repeatability in stress responses when compared to the wild-caught group, with fish selectively bred for a high-stress coping style having the least variation in stress responses across time [31]. This observation highlights the potential for anxiety-related behaviours to exhibit consistent, trait-like stability across time and contexts, though further research is needed to better understand the relative contributions of individual variation and environmental influences across longer timeframes.

The present study focused on characterizing patterns of stability and variability in anxiety-like behaviours in zebrafish at both the group and individual levels over a 21-week period. Fish were tested in the novel tank dive test at 90, 120, and 150 days post-fertilization, with individual identities tracked across time. Age- and sex-specific differences emerged in locomotor activity and zone preference. By quantifying behavioural consistency and change over development, this study provides a framework for identifying stable phenotypes and advancing selective breeding approaches in zebrafish models of anxiety.

## 2. Methods

### 2.1. Animals and housing

Adult AB/wild-type hybrid zebrafish of mixed sex were obtained from MacEwan University's in-house breeding facility (Edmonton, Alberta, Canada). Fish were originally sourced from Dalhousie University (Halifax, NS) and the University of Ottawa (Canada) and were of AB and unknown wildtype strains, respectively. At the MacEwan facility, a new strain of AB is introduced to the population from Dalhousie every 2–3 years to maintain the hybrid line. All zebrafish were housed in a Tecniplast ZebTEC multilinking system (Tecniplast Group, Toronto ON, Canada). To identify individuals during the longitudinal testing period, zebrafish were housed in pairs (1:1 male to female) in 3.5 L polycarbonate tanks separated by transparent, perforated polycarbonate inserts, with each compartment containing plastic plant and floor enrichments (Fig 1A). While fish were physically separated to enable individual identification across repeated behavioural sessions, the dividers allowed continuous visual and olfactory contact. This housing strategy aligns with evidence that sensory access to conspecifics mitigates the behavioural effects of partial isolation in zebrafish [32]. To mitigate eggbinding, spawning substrates were placed in each tank to facilitate natural egg release. In a few cases where females appeared egg-bound, they were removed from the dataset prior to any intervention. Housing facility water was continuously re-circulated through the habitat by automated water exchange and treated by a 5-stage filtration system. Water temperature was maintained at 27 °C and pH was maintained between 6.5 and 8.0. Zebrafish were on a 14-hour light/10-hour dark photoperiod from 7:00 AM to 9:00 PM and fed twice daily with Gemma Micro fish pellets for adults (GEMMA Micro, Maine, USA). Sex was identified based on secondary sexual characteristics, including body shape and colouration. Throughout the 21-week study, 2 males and 7 females died, either due to natural causes (e.g., senescence, egg binding) or were humanely euthanized following signs of poor health. Mortality occurred sporadically and not at specific timepoints. All data from deceased fish were completely removed from the analysis. To prevent social isolation of the remaining fish, a non-experimental fish was introduced into the adjacent compartment; these replacements were not included in any behavioural testing or analysis. Therefore, the results presented here are based on n = 48 male and n = 43 female zebrafish for a total sample size of N = 91. In addition to the longitudinal cohort, a separate naïve cohort of AB/wild-type hybrid zebrafish was used to control for potential effects of repeated testing. These fish were drawn from the same breeding facility and maintained and tested under the same conditions as described, but were tested only once at either 90, 120, or 150 dpf. The naïve cohort consisted of n = 20 fish per age group (balanced for sex), and did not undergo any behavioural testing prior to their single Novel Tank Dive Test exposure.

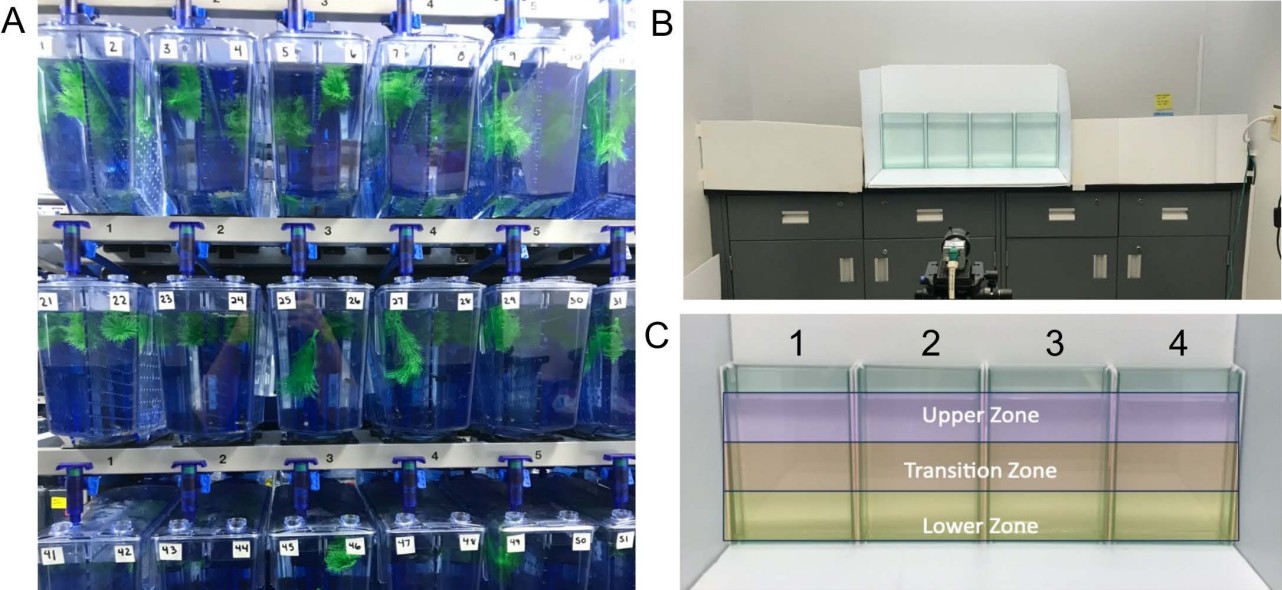

**Fig 1. Housing and behavioural testing.** A) Zebrafish (N = 100) were housed in pairs (1 female:1 male) and individually tested over a 21 week period to assess locomotor and anxiety-like behaviour at several adult stages (90, 120, and 150 dpf). B) The multiple-arena modified novel tank dive arena dimensions allowed for a deeper water level when compared to traditional arenas, as well as a narrower tank-depth to reduce 3D exploratory behaviour. Swimming behaviour was tracked and recorded to identify the potential trait-like characteristics of anxiety-like and locomotor behaviour. C) Virtual zones created within EthoVision: the upper zone, the transition zone, and the lower zone.

## 2.2. Behaviour testing

### 2.2.1. Adult zebrafish behavioural testing.
Longitudinal data collection took place over 21 weeks at 90, 120, and 150 days post fertilization (dpf). Behavioural testing followed previously described methods [33]. Briefly, on testing days zebrafish were transferred in their habitat tanks to the adjacent testing room and acclimated for a minimum of 15 minutes prior to testing. The habitat tanks and testing arena were fully surrounded by white corrugated plastic to reduce extraneous visual stimuli. After acclimation, individual zebrafish were transferred directly into the testing arena where behaviour was tracked and recorded for 10 minutes. To minimize positional and time-of-day effects, fish were tested in four identical arenas with testing order rotated across developmental timepoints, ensuring that each individual was assessed at different times of day and in different arena positions across sessions. Arena water was changed between trials to prevent build-up of waste and to maintain water temperature. After each trial, zebrafish were returned to their habitat tank and fed. The water temperature in the habitat tanks and testing arena was maintained between 26–28 °C with seedling heat mats (Hydrofarm Horticultural Products, Petaluma CA). Luminance in the testing room was ~ 32 cd/m² (cal SPOT photometer; Cooke Corp. CA, USA). Zebrafish swimming behaviour was recorded by a Basler GenICam acA1300-60gc Area Scan video camera (Basler Inc., USA) at 25 frames per second, mounted to a tripod. All zebrafish were tested in an identical manner and swimming behaviour was tracked and recorded using Noldus EthoVision XT ® tracking software (v. 17.0, Noldus, Wageningen, NL).

## 2.3. The novel tank dive test

The novel tank dive test measures vertical exploration and locomotor behaviour. Vertical exploration was quantified using two measures: zone occupancy, calculated as the cumulative time spent in the upper, transition, and lower zones, and distance from bottom, defined as the mean vertical position of each fish across the 10-minute trial. Locomotor behaviour was

quantified as swimming velocity and time spent immobile, with immobility defined as less than a 5% change in the pixels of the body of the fish from one sample to the next. The modified novel tank dive test, described in Johnson et al. [33], used glass arenas measuring 25 × 18 × 5.5 cm (height × width × depth) filled to a water level of 20 cm (1.4 L; Fig 1B). Arenas were divided into three virtual zones within EthoVision XT (upper, transition, lower; Fig 1C). Four identical arenas were used to track four fish at a time with the multiple-arena module on EthoVision XT.

## 2.4. Statistical analysis

For each behaviour variable, data distributions were assessed using the D'Agostino-Pearson omnibus normality test. Because repeated measures across time within subjects violated normality assumptions, age-related differences were evaluated using the non-parametric Friedman test, followed by Dunn's multiple comparisons test to identify significant pairwise differences across timepoints. To assess sex differences and age × sex interactions, a two-way ANOVA was conducted for each behavioural measure, treating age and sex as between-subjects factors. Post hoc comparisons were performed using Fisher's Least Significant Difference (LSD) test. These analyses did not account for within-subject matching and were treated as independent group comparisons across age and sex. Although the data were non-normally distributed, ANOVAs were applied due to their robustness to moderate deviations from normality in balanced designs with large sample sizes. To assess potential effects of repeated exposure, naïve and longitudinal groups were compared at each developmental stage (90, 120, 150 dpf). Because Bartlett's test indicated unequal variances, Brown–Forsythe ANOVAs were used for these comparisons, followed by Dunnett's T3 post hoc test. An alpha level of $p < .05$ and 95% confidence intervals were used to assess statistical significance. All values are presented as mean ± standard error of the mean (S.E.M.).

To examine sources of variation in each trait, we developed a series of univariate models using Gaussian error distributions using the 'lmer' function from the 'lme4' package in the R statistical environment. Residual errors were assessed for normality and met this assumption when traits were analyzed on the observed scale for time spent in the lower zone and average velocity. For total time spent immobile, normality of residuals was achieved following log transformation. Separate univariate models were constructed for each sex and time period (i.e., early period = 90–120 dpf, late period = 120–150 dpf). Age was included as a fixed effect in all models to evaluate the effect of age on mean trait expression, and fish ID was included as a random effect to compare within- and among-individual variance in traits across the age and sex-specific datasets. Adjusted repeatabilities (i.e., repeatability after correcting for age) were calculated following Nakagawa and Schielzeth [34]. To study age-dependent changes in repeatability, the change in repeatability for each trait between period 1 and period 2 was compared. Comparisons between sexes for the same time period are also be reported. We used the 'sim' function of the 'arm' package [35] to simulate values of the posterior distribution of the model parameters [36]. This function applies flat, uninformative priors to lmer estimates, allowing posterior distributions of parameters to be generated and 95% credible intervals (CIs) to be calculated. These CIs provide a way to directly evaluate the extent of overlap between estimates, which in turn indicates both whether differences are supported and the magnitude of those differences. We then extracted 95% credible intervals (CIs) around the mean (β) based on 1,000 simulations using the MCMCglmm package [37]. The 95% CI indicates a margin of error in terms of a range of plausible values for β. With this CI, we indicate that we are 95% confident that our CI includes the actual effect size [38]. To evaluate the differences between means and CIrs of the fixed effects, as well as the variance components and the repeatability estimates, we followed Cumming and Finch [38]. Independent 95% Crs were deemed to indicate significant differences between averages when they did not overlap. This corresponds to a traditional p value of <0.006. In cases where 95% credible intervals (CIs) overlapped, we estimated the proportion of overlap by dividing the intersecting range by the average width of the two intervals [39]. This proportion was used to characterize the strength of evidence for differences between parameters as a graded, continuous measure rather than a binary significant/non-significant outcome. The proportion overlap between the 95% CI of two independent estimates provides a conservative estimate of p-value [39], and thus, effects whose pr < 0.05 are also described as being significantly different.

 

A priori power analysis was conducted using G*Power (v3.1.9.7) [40] to evaluate the adequacy of the sample size. For the repeated-measures analyses across three timepoints, assuming a moderate correlation among repeated measures (r = 0.5), α = 0.05, and a medium effect size (f = 0.25), the sample size of n = 100 (50 males, 50 females) yielded a power of 0.87. For between-group comparisons by sex, using α = 0.05 and a medium effect size (Cohen's d = 0.5), the power was 0.80. These estimates indicate that the study was sufficiently powered to detect medium-sized effects both within and between subjects.

## 2.5. Ethics statement

All experiments were approved by the University of Alberta Animal Care and Use Committee (ACUC) under protocol number 00004663 and MacEwan University Animal Ethics Board (AREB) under protocol number 101853 in compliance with the Canadian Council for Animal Care (CCAC) experimental guidelines. This study was also carried out in compliance with ARRIVE guidelines for animal research.

## 3. Results

### 3.1. Zone preference

On average, zebrafish spent significantly more time in the upper zone (H(2) = 25.13, p < 0.0001) at 150 dpf (219.5 ± 11.12 s) when compared to 90 (145.8 ± 11.27 s, p < 0.0001) and 120 dpf (177.5 ± 13.59 s, p = 0.0015; Fig 2A). Zebrafish also spent significantly less time in the lower zone (H(2) = 14.0, p = 0.0009) at 150 dpf (240.2 ± 12.96 s) when compared to 90 (326.5 ± 15.59 s, p = 0.0007) and 120 dpf (290.9 ± 16.72 s, p = 0.0433; Fig 2B). These results suggest zebrafish spend less time in the lower zone as they age. Consistent with these zone-based measures, distance from bottom also showed a significant effect of age (H(2) = 14.0, p = 0.0009), with zebrafish at 150 dpf positioned higher in the water column than at 90 (p = 0.0008) and 120 dpf (p = 0.028), whereas 90 and 120 dpf did not differ (p = 0.898; Fig 2C). However, distance from bottom did not reveal additional differences beyond those detected with lower zone occupancy. These results could also suggest that zebrafish become more familiar with the novel testing environment over repeated exposure. To evaluate whether repeated exposure influenced behaviour, a naïve cohort of zebrafish tested only once at 90, 120, or 150 dpf was compared to the longitudinally tested cohort. A Brown–Forsythe ANOVA indicated a significant overall effect of group (F(5, 120.6) = 3.92, p = 0.0025), but Dunnett's T3 post hoc tests revealed no significant differences in time spent in the lower zone between naïve and longitudinal fish at any age (all p > 0.99; Fig 2F). This suggests that repeated testing did not confound age-related behavioural trends.

### 3.2. Locomotion

On average, zebrafish swimming velocity significantly increased (H(2) = 15.63, p = 0.0004) at 150 dpf (6.73 ± 0.23 cm/s) when compared to 90 (6.34 ± 0.29 cm/s, p = 0.0115) and 120 dpf (5.83 ± 0.24 cm/s, p = 0.0005; Fig 2D). Zebrafish also spent less time immobile (H(2) = 7.670, p = 0.0216) at 150 dpf (48.86 ± 9.13 s) when compared to 120 dpf (67.72 ± 11.17 s, p = 0.0183; Fig 2E).

### 3.3. Sex differences

There were no significant overall sex differences in time spent in the upper zone (F(1, 267) = 0.7647, p = 0.3827; Fig 3A) or lower zone (F(1, 267) = 1.757, p = 0.1861; Fig 3B). However, within-group changes over time were observed. In females, time spent in the upper zone increased significantly at 150 dpf compared to 90 dpf (p = 0.0026; Fig 3A). In males, upper zone time increased at 150 dpf compared to both 90 (p = 0.0025) and 120 dpf (p = 0.0442), with a significant main effect of age (F(2, 267) = 9.335, p = 0.0001; Fig 3A). Time spent in the lower zone also decreased across timepoints for both sexes. Females spent significantly less time in the lower zone at 150 dpf compared to 90 dpf (p = 0.0259; Fig 3B), while males spent less time in the lower zone at 150 dpf compared to both 90 (p = 0.0020) and 120 dpf (p = 0.0321), with a

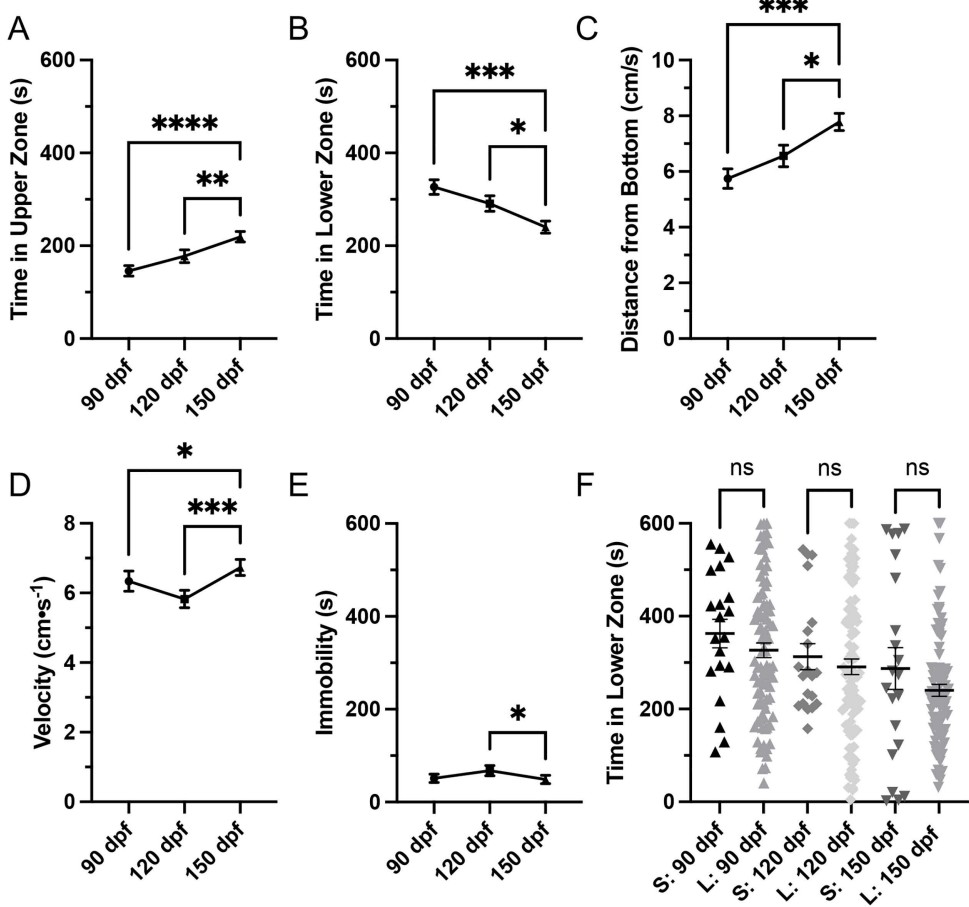

**Fig 2. Zone preference and locomotion.** A) Zebrafish spent significantly more time in the upper zone at 150 dpf when compared to 90 and 120 dpf and B) less time in the lower zone at 150 dpf when compared to 90 and 120 dpf. C) Distance from bottom also differed with age, with fish at 150 dpf positioned higher in the water column compared to 90 and 120 dpf. D) Zebrafish swimming velocity significantly increased at 150 dpf when compared to 90 and 120 dpf. E) Zebrafish also spent less time immobile at 150 dpf when compared to 120 dpf. F) Single-exposure cohort (S) compared to longitudinal cohort (L). Naïve zebrafish tested only once at 90, 120, or 150 dpf did not differ from the longitudinal zebrafish used in this study repeatedly tested across time in time spent in the lower zone, indicating repeated exposure did not alter behaviour. All data are presented as mean ± S.E.M. Significant differences are indicated by * (p < 0.05), *** (p < 0.001), and **** (p < 0.0001).

significant main effect of age (F(2, 267) = 7.132, p = 0.0010; Fig 3B). These results indicate that while both sexes exhibited age-related behavioural changes, there were no significant differences between sexes. There were significant sex differences in swimming velocity (F(1, 267) = 43.15, p < 0.0001). Males had higher swimming speed at 90 (p = 0.0001), 120 (p = 0.0044), and 150 dpf (p < 0.0001) when compared to females. Interestingly, females swimming velocity did not change across time, whereas males swimming velocity increased over time (F(2, 267) = 3.382, p = 0.0354; Fig 3C). There were no significant sex differences in time spent immobile (F(1, 267) = 3.448, p = 0.0644), and immobility did not change across time for either males or females (F(2, 267) = 1.059, p = 0.3484; Fig 3D).

### 3.4. Individual variation

The proportion of time spent in the lower zone of the tank decreased with age in both males and females, and across both the first (90–120 dpf) and second (120–150 dpf) observation periods. For males, time spent in the lower zone decreased

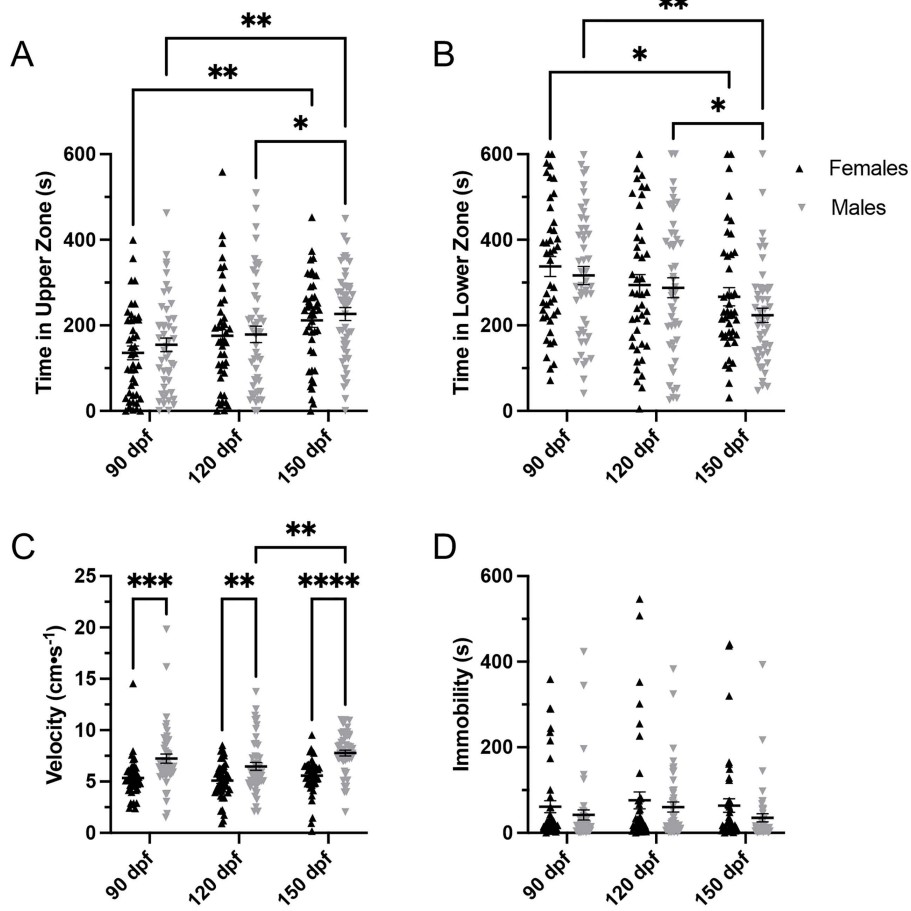

**Fig 3. Sex differences.** A) There were no significant sex differences in time spent in the upper zone. However, females spent more time in the upper zone at 150 dpf compared to 90 dpf, and males spent more time in the upper zone at 150 dpf compared to both 90 and 120 dpf. B) There were no significant sex differences in time spent in the lower zone. However, females spent less time in the lower zone at 150 dpf compared to 90 dpf, and males spent less time in the lower zone at 150 dpf compared to both 90 and 120 dpf. C) There were significant sex differences in swimming velocity. Males had higher swimming speed at 90, 120, and 150 dpf compared to females. Interestingly, females' swimming velocity did not change over time, whereas males' swimming velocity increased over time. D) There were no sex differences in immobility. Both males' and females' time spent immobile did not change over time. All data are presented as mean±S.E.M. Significant differences are indicated by * (p<0.05), ** (p<0.01), *** (p<0.001), and **** (p<0.0001).

significantly over the first period (β=−53.29, 95% CI [−90.27, −27.55], pr=0.08), with an even greater decrease during the later observation period (β=−78.14, 95% CI [−94.79, −18.98], pr=0.00). Females showed a similar age-related decline in time spent in the lower zone, with a significant reduction in the first period (β=−49.16, 95% CI [−75.47, 20.47], pr=0.09) but a lesser decrease during the later period (β=−28.58, 95% CI [−37.65, −11.86], pr=0.00; Table 1).

In contrast, the age-related changes in velocity and log time immobile were non-linear. Males showed a decrease in swimming velocity with age in the first period (β=−0.51, 95% CI [−1.90, 0.03], pr=0.04) and an increase in the second period (β=1.37, 95% CI [0.78, 1.89], pr=0.00). Females displayed a similar pattern, with minimal age effects during the first period (β=−0.05, 95% CI [−0.98, 0.26], pr=0.30) but a significant increase during the late period (β=0.55, 95% CI [−0.01, 1.04], pr=0.02; Table 2). For log time spent immobile, males demonstrated an increase in immobility in the first period (β=0.45, 95% CI [−0.08, 0.92], pr=0.06) and a decrease in the second period (β=−0.59, 95% CI [−1.04, −0.21], pr=0.003). Females exhibited smaller and non-significant changes across both periods (e.g., β=−0.06, 95% CI [−0.33,

**Table 1. Sources of variation in time spent in the lower zone of the tank (in seconds) over a 10-minute observation.**

| | Males | | | Females | | |
|---|---|---|---|---|---|---|
| | Overall | Early | Late | Overall | Early | Late |
| **Fixed effects** | β±95% CI | β±95% CI | β±95% CI | β±95% CI | β±95% CI | β±95% CI |
| Intercept[1] | 331.42 (280.04, 359.54) | 306.39 (269.35, 359.73) | 362.59 (273.75, 420.81) | 334.02 (297.95, 379.67) | 333.20 (297.53, 385.03) | 337.95 (278.99, 386.22) |
| Age | −47.64 (−74.76, −24.06) pr=0.00 | −53.29 (−90.27, 27.55) pr=0.08 | −78.14 (−94.79, −18.98) pr=0.00 | −36.20 (−59.13, −20.72) pr=0.00 | −49.16 (−75.47, 20.47) pr=0.09 | −28.58 (−37.65, −11.86) pr=0.00 |
| **Random effects** | σ±95% CI | σ±95% CI | σ±95% CI | σ±95% CI | σ±95% CI | σ±95% CI |
| FishID | 5658.59 (4150.60, 8173.51) | 5414.16 (3173.71, 7308.30) | 9691.39 (7146.86, 13516.62) | 9566.37 (7241.29, 12855.35) | 8768.88 (5030.92, 10728.49) | 15110.53 (12782.17, 20090.68) |
| Residual | 15121.89 (11683.27, 17126.76) | 19870.45 (14533.57, 25094.16) | 10444.16 (8097.24, 14404.00) | 11445.73 (9334.08, 14948.49) | 13771.98 (11853.43, 21334.97) | 5314.23 (4013.99, 7082.14) |
| **Repeatability** | r±95% CI | r±95% CI | r±95% CI | r±95% CI | r±95% CI | r±95% CI |
| FishID | 0.29 (0.22, 0.36) | 0.25 (0.14, 0.27) | 0.48 (0.41, 0.56) | 0.43 (0.36, 0.52) | 0.33 (0.25, 0.42) | 0.75 (0.70, 0.82) |

Values presented are the posterior modes (β, σ or r) and 95% credible intervals. The proportion of overlap with zero (pr) for effects of age are also provided to aid in interpretation of the strength of support for an effect in cases where the 95% CrI overlaps with zero. Note, random effects are constrained to be ≥ 0, so proportion overlap with zero is not meaningful.

[1]Intercept estimated for 90 dpf, the first age at which observations were made.

[2]Age effect is for each 30 days increase in age, which corresponds to the observation interval in the study.

**Table 2. Sources of variation in velocity (cm/s) of zebrafish over a 10-minute observation.**

| | Males | | | Females | | |
|---|---|---|---|---|---|---|
| | Overall | Early | Late | Overall | Early | Late |
| **Fixed effects** | β±95% CI | β±95% CI | β±95% CI | β±95% CI | β±95% CI | β±95% CI |
| Intercept | 7.10 (6.02, 7.55) | 7.36 (6.47, 8.00) | 5.14 (4.17, 6.21) | 5.06 (4.62, 5.66) | 5.35 (4.90, 5.99) | 4.59 (3.62, 5.31) |
| Age | 0.44 (−0.11, 0.72) pr=0.10 | −0.51 (−1.90, 0.03) pr=0.04 | 1.37 (0.78, 1.89) pr=0.00 | 0.11 (−0.12, 0.47) pr=0.28 | −0.05 (−0.98, 0.26) pr=0.30 | 0.55 (−0.01, 1.04) pr=0.02 |
| **Random effects** | σ±95% CI | σ±95% CI | σ±95% CI | σ±95% CI | σ±95% CI | σ±95% CI |
| FishID | 1.89 (1.15, 2.37) | 1.21 (0.68, 1.74) | 2.90 (2.28, 4.08) | 1.09 (0.78, 1.53) | 0.81 (0.43, 1.18) | 1.61 (1.02, 2.00) |
| Residual | 5.86 (4.28, 6.30) | 6.85 (5.17, 9.39) | 2.24 (1.78, 3.17) | 2.31 (1.84, 3.02) | 2.93 (2.35, 3.91) | 1.78 (1.36, 2.43) |
| **Repeatability** | r±95% CI | r±95% CI | r±95% CI | r±95% CI | r±95% CI | r±95% CI |
| FishID | 0.25 (0.18, 0.31) | 0.16 (0.10, 0.21) | 0.56 (0.47, 0.62) | 0.32 (0.24, 0.38) | 0.21 (0.14, 0.27) | 0.49 (0.39, 0.56) |

Values presented are the posterior modes (β, σ or r) and 95% credible intervals. The proportion of overlap with zero (pr) for effects of age are also provided to aid in interpretation of the strength of support for an effect in cases where the 95% CrI overlaps with zero. Note, random effects are constrained to be ≥ 0, so proportion overlap with zero is not meaningful.

[1]Intercept estimated for 90 dpf, the first age at which observations were made.

[2]Age effect is for each 30 days increase in age, which corresponds to the observation interval in the study.

0.13], pr=0.26 overall). Age-related changes in immobility were smaller in absolute magnitude in females (as shown by a greater overlap with zero (Table 3).

Among-individual variance was lower in males for time spent in the lower zone (σ=5658.59, 95% CI [4150.60, 8173.51] for males vs. σ=9566.37, 95% CI [7241.29, 12855.35] for females) but increased with age in both males and females

**Table 3. Sources of variation in time spent immobile (seconds) in zebrafish over a 10-minute observation.**

| | Males | | | Females | | |
|---|---|---|---|---|---|---|
| | Overall | Early | Late | Overall | Early | Late |
| **Fixed effects** | β±95% CI | β±95% CI | β±95% CI | β±95% CI | β±95% CI | β±95% CI |
| Intercept[1] | 2.92 (2.57, 3.33) | 2.79 (2.44, 3.17) | 3.83 (3.11, 4.39) | 3.39(2.84, 3.62) | 3.25 (2.82, 3.63) | 3.24 (2.51, 3.84) |
| Age[2] | −0.11 (−0.32, 0.08) pr=0.12 | 0.45 (−0.08, 0.92) pr=0.06 | −0.59 (−1.04, −0.21) pr=0.003 | −0.06 (−0.33, 0.13) pr=0.26 | 0.12 (−0.34, 0.60) pr=0.42 | −0.05 (−0.50, 0.25) pr=0.28 |
| **Random effects** | σ±95% CI | σ±95% CI | σ±95% CI | σ±95% CI | σ±95% CI | σ±95% CI |
| FishID | 0.73 (0.51, 0.97) | 0.54 (0.40, 0.86) | 0.95 (0.75, 1.39) | 1.09 (0.79, 1.40) | 0.75 (0.50, 1.13) | 1.54 (1.17, 1.97) |
| Residual | 1.52 (1.34, 1.84) | 1.61 (1.00, 2.03) | 1.14 (0.85, 1.47) | 0.90 (0.86, 1.30) | 1.17 (0.85, 1.64) | 0.73 (0.59, 1.08) |
| **Repeatability** | r±95% CI | r±95% CI | r±95% CI | r±95% CI | r±95% CI | r±95% CI |
| FishID | 0.37 (0.27, 0.41) | 0.27 (0.20, 0.36) | 0.51 (0.37, 0.54) | 0.51 (0.44, 0.58) | 0.41 (0.30, 0.52) | 0.65 (0.56, 0.73) |

Data were log transformed prior to analyses. Values presented are the posterior modes (β, σ or r) and 95% credible intervals. The proportion of overlap with zero (pr) for effects of age are also provided to aid in interpretation of the strength of support for an effect in cases where the 95% CrI overlaps with zero. Note, random effects are constrained to be ≥ 0, so proportion overlap with zero is not meaningful.

[1]Intercept estimated for 90 dpf, the first age at which observations were made.

[2]Age effect is for each 30 days increase in age, which corresponds to the observation interval in the study.

(σ = 5414.16, 95% CI [3173.71, 7308.30] in the early period vs. σ = 9691.39, 95% CI [7146.86, 13516.62] in the late period for males, and σ = 8768.88, 95% CI [5030.92, 10728.49] in the early period vs. σ = 15110.53, 95% CI [12782.17, 20090.68] in the late period for females; Table 1; Fig 4A). At the same time, males exhibited higher within-individual (residual) variance for time spent in the lower zone (σ = 15121.89, 95% CI [11683.27, 17126.76] for males vs. (σ = 11445.73, 95% CI [9334.08, 14948.49] for females) which decreased with age (σ = 19870.45, 95% CI [14533.57, 25094.16] in the early period vs. σ = 10444.16, 95% CI [8097.24, 14404.00] in the late period for males, and σ = 13771.98, 95% CI [11853.43, 21334.97] in the early period vs. σ = 5314.23, 95% CI [4013.99, 7082.14] in the late period for females; Table 1; Fig 4B). Repeatability increased with age for time spent in the lower zone and females exhibited higher repeatability than males at all ages (r = 0.25, 95% CI [0.14, 0.27] in the early period vs. r = 0.48, 95% CI [0.41, 0.56] in the late period for males, and r = 0.33, 95% CI [0.25, 0.42] in the early period vs. r = 0.75, 95% CI [0.70, 0.82] in the late period for females; Table 1 Fig 4C).

Among-individual variability in velocity was greater for males than for females overall (σ = 1.89, 95% CI [1.15, 2.37] for males vs. σ = 1.09, 95% CI [0.78, 1.53] for females). This variability increased with age in both sexes, with males showing an increase from σ = 1.21 (95% CI [0.68, 1.74]) in the early period to σ = 2.90 (95% CI [2.28, 4.08]) in the late period, and females increasing from σ = 0.81 (95% CI [0.43, 1.18]) in the early period to σ = 1.61 (95% CI [1.02, 2.00]) in the late period (Table 2; Fig 4D). In contrast, males showed higher within-individual variation in velocity compared to females (σ = 5.86, 95% CI [4.28, 6.30] for males vs. σ = 2.31, 95% CI [1.84, 3.02] for females). This variation decreased with age in both sexes, with males showing a reduction from σ = 6.85 (95% CI [5.17, 9.39]) in the early period to σ = 2.24 (95% CI [1.78, 3.17]) in the late period, and females decreasing from σ = 2.93 (95% CI [2.35, 3.91]) in the early period to σ = 1.78 (95% CI [1.36, 2.43]) in the late period (Table 2; Fig 4E). Velocity repeatability increased with age for both sexes and was higher for females compared to males. For example, repeatability increased from r = 0.16 (95% CI [0.10, 0.21]) in the early period to r = 0.56 (95% CI [0.47, 0.62]) in the late period for males, and from r = 0.21 (95% CI [0.14, 0.27]) in the early period to r = 0.49 (95% CI [0.39, 0.56]) in the late period for females (Table 2; Fig 4F).

Males showed lower variability in immobility than females overall (σ = 0.73, 95% CI [0.51, 0.97] for males vs. σ = 1.09, 95% CI [0.79, 1.40] for females). This variability increased with age, rising from σ = 0.54 (95% CI [0.40, 0.86]) in the early period to σ = 0.95 (95% CI [0.75, 1.39]) in the late period for males, and from σ = 0.75 (95% CI [0.50, 1.13]) in the early period to σ = 1.54 (95% CI [1.17, 1.97]) in the late period for females (Table 3; Fig 4G). Within-individual variability in immobility was higher in males than females (σ = 1.52, 95% CI [1.34, 1.84] for males vs.

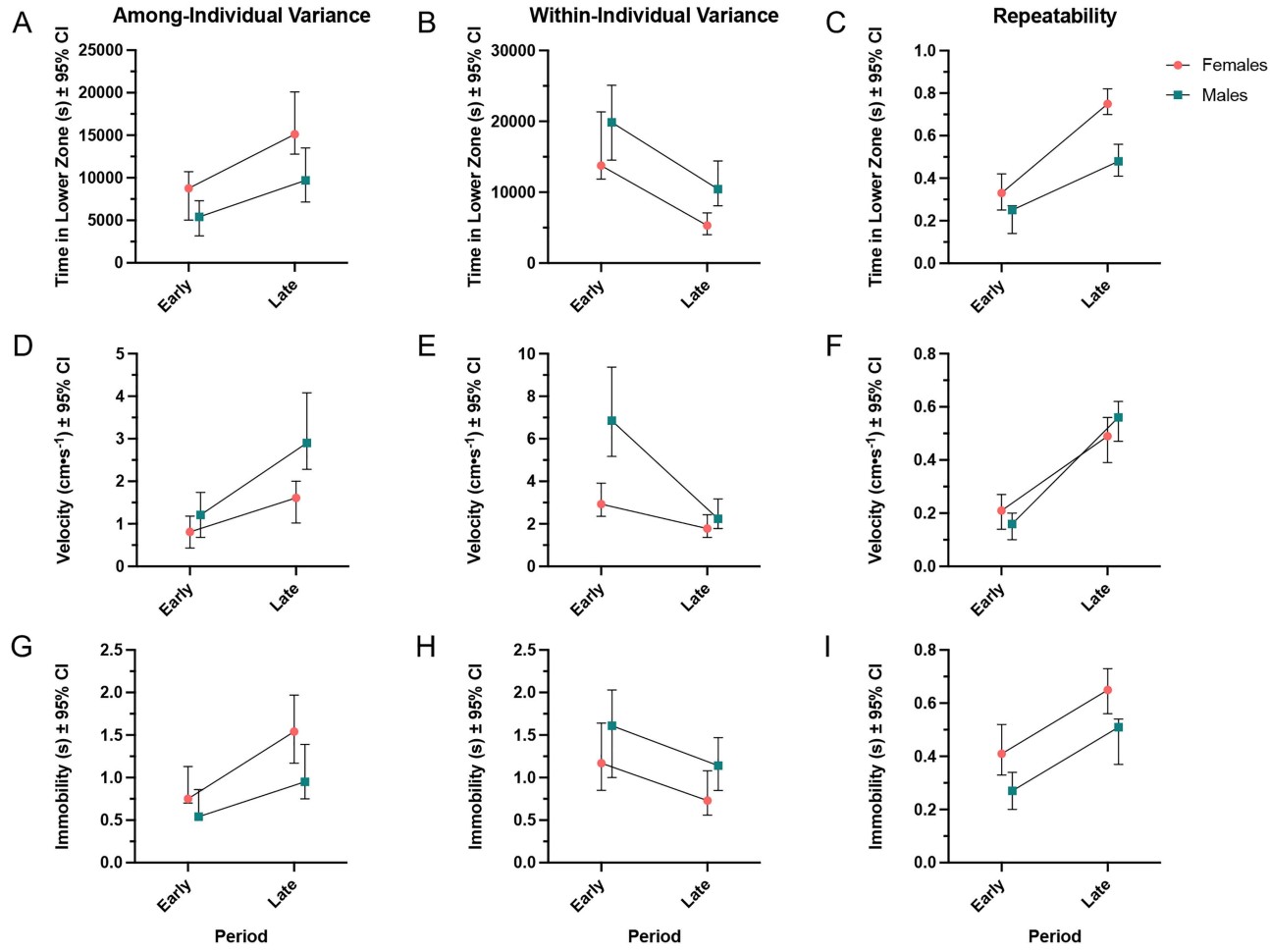

**Fig 4. Sex- and age-related changes in individual variance and behavioural repeatability.** Model-derived estimates (± 95% credible intervals) of among-individual variance, within-individual (residual) variance, and repeatability for each behavioural trait, separated by sex and age group (Early = 90–120 dpf; Late = 120–150 dpf). Note, y-axis labels provide the description (and units) of the response variable to which the variance component, though it should be noted that variance components themselves, including repeatability, are unitless. A) Among-individual variance in time spent in the lower zone was higher in females than males and increased with age in both sexes. B) Within-individual variance in lower zone time was higher in males and decreased with age in both sexes. C) Repeatability for lower zone time increased with age and was higher in females. D) Among-individual variance in velocity was higher in males and increased with age in both sexes. E) Within-individual variance in velocity was higher in males and decreased with age in both sexes. F) Velocity repeatability increased with age and was higher in females in the early period and higher in males in the later period. G) Among-individual variance in immobility was higher in females and increased with age in both sexes. H) Within-individual variance in immobility was higher in males and decreased with age in both sexes. I) Immobility repeatability increased with age and was higher in females than males. All data are presented as point estimates with error bars representing 95% confidence intervals.

$\sigma = 0.90$, 95% CI [0.86, 1.30] for females) and decreased with age. Males exhibited a decline in within-individual variation from $\sigma = 1.61$ (95% CI [1.00, 2.03]) in the early period to $\sigma = 1.14$ (95% CI [0.85, 1.47]) in the late period, while females decreased from $\sigma = 1.17$ (95% CI [0.85, 1.64]) in the early period to $\sigma = 0.73$ (95% CI [0.59, 1.08]) in the late period (Table 3; Fig 4H). Repeatability for immobility increased with age and was generally higher in females than in males. Repeatability rose from $r = 0.27$ (95% CI [0.20, 0.36]) in the early period to $r = 0.51$ (95% CI [0.37, 0.54]) in the late period for males, and from $r = 0.41$ (95% CI [0.30, 0.52]) in the early period to $r = 0.65$ (95% CI [0.56, 0.73]) in the late period for females (Table 3; Fig 4I).

## 3.5. Group variation

To determine whether distinct high, medium, and low anxiety-level groups could be established based on their zone preference, we performed post-hoc rankings based on the cumulative time each fish (N = 91) spent in the bottom zone of the arena at 150 dpf. The 10 male and female zebrafish who spent the most time in the bottom zone were labelled high anxiety (HAZ), while the 10 zebrafish who spent the least time in the bottom zone were labelled low anxiety (LAZ) and the 10 zebrafish falling in the middle were labelled medium anxiety fish (MAZ). Their group behaviour was then analyzed at 90 and 120 dpf in comparison to 150 dpf to determine whether the assigned anxiety-level was stable over time within and between groups. For within group differences, there were no significant changes in time spent in the upper zone at 90, 120, and 150 dpf in the HAZ and MAZ groups, however, the LAZ group spent significantly more time in the upper zone at 120 and 150 dpf ($F_{(2, 81)}$ = 5.065, p = 0.0085; Fig 5A). There were also no significant changes in time spent in the lower zone in the HAZ and MAZ groups, however, the LAZ group spent significantly less time in the lower zones at 120 and 150 dpf ($F_{(2, 81)}$ = 4.734, p = 0.0114; Fig 5B).

Fish in the HAZ group consistently spent significantly less time in the upper zone than the MAZ and LAZ groups at 90, 120, and 150 dpf. The LAZ group spent the same amount of time in the upper zone as the MAZ group at 90 dpf, but spent significantly more time in the upper zone at 120 and 150 dpf ($F_{(2, 81)}$ = 58.71, p < 0.0001; Fig 5C). Similarly, fish in the HAZ group consistently spent significantly more time in the lower zone than the MAZ and LAZ groups at 90, 120, and 150 dpf. The LAZ group spent the same amount of time in the lower zone as the MAZ group at 90 dpf, but spent significantly less time in the lower zone at 120 and 150 dpf ($F_{(2, 81)}$ = 79.13, p < 0.0001; Fig 5D).

## 4. Discussion

Here we examined the stability and variability of zone preference and locomotion behaviours in zebrafish. Zebrafish exhibited age-related increases in time spent in the upper portion of the testing arena and swimming velocity, accompanied by decreases in anxiety-like behaviours reflected by reduced time spent in the lower zone and time spent immobile. Sex differences in locomotor activity emerged with age, with males showing higher variability and greater increases in velocity over time, while females displayed greater behavioural stability. Additionally, behavioural profiles remained stable across time for high-, medium-, and low-anxiety groups, suggesting trait-like characteristics of anxiety in zebrafish. These findings emphasize the importance of accounting for age, sex, and individual variation in behavioural and pharmacological research.

### 4.1. Anxiety-like and locomotor variables

Zebrafish exhibited a significant age-related increase in time spent in the upper zone and a corresponding decrease in time spent in the lower zone of the tank. This trend may reflect an inherent reduction in anxiety-like behaviour with age. Comparisons with a separate naïve cohort tested only once at each age confirmed that repeated exposure did not drive the observed age-related shifts in vertical exploration. Naïve and longitudinal cohorts did not differ in lower-zone time at any age, indicating that the developmental trends reported here are unlikely to reflect habituation to the test. We also analyzed distance from bottom as a continuous measure of vertical exploration. This metric produced results consistent with the zone-based analysis, showing that zebrafish at 150 dpf were positioned higher in the water column compared to 90 and 120 dpf, but it did not reveal additional differences beyond those detected with zone occupancy. This supports the validity of zone occupancy as a robust and interpretable indicator of anxiety-like behaviour, while acknowledging that continuous measures may be useful in combination with other behavioural metrics in future work. Previous studies have highlighted that time spent in the lower zone of the tank is a robust indicator of anxiety in zebrafish [1]. In our study, this measure showed clear age-related shifts, our results challenge the assumption that this behaviour remains static over time, demonstrating that age must be accounted for when interpreting anxiety-like behaviours. Decreased time spent in the lower zone at 150 dpf aligns with findings by Dereje et al. [23], who reported that zebrafish strains displaying lower

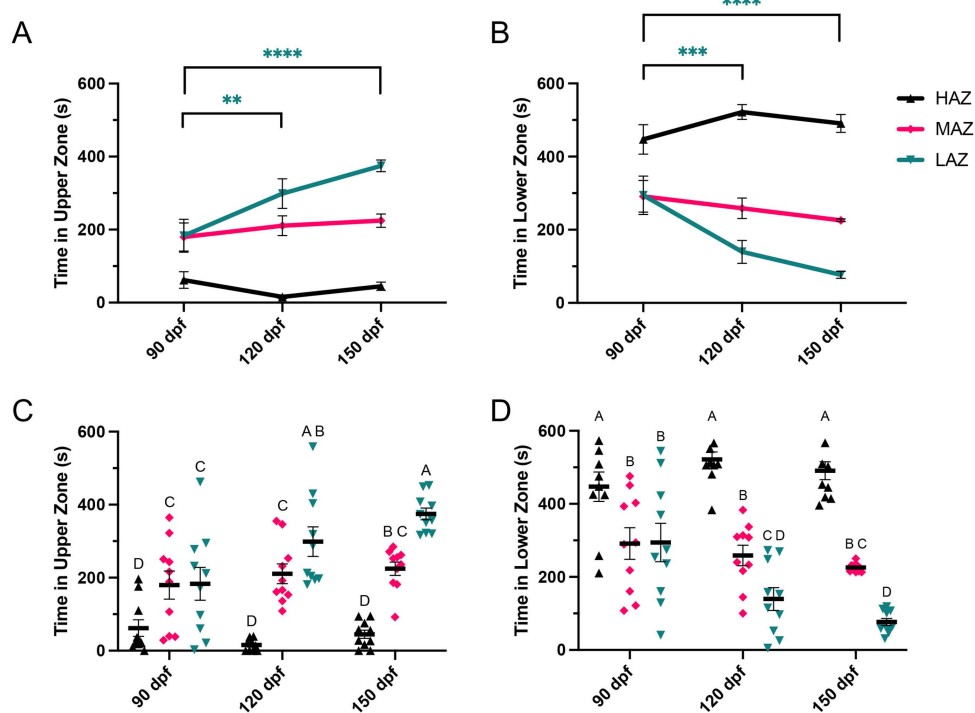

**Fig 5. Group variation.** A) There were no differences in time spent in the upper zone in the high (HAZ) or medium (MAZ) anxiety groups across time. Fish in the low (LAZ) anxiety group spent more time in the upper zone at 120 and 150 days old compared to 90 days old. B) There were also no differences in time spent in the lower zone in the HAZ or MAZ groups across time. Fish in the LAZ group spent less time in the lower zone at 120 and 150 days old compared to 90 days old. C) Fish in the HAZ group spent less time in the upper zone compared to the medium and low anxiety groups at 90, 120, and 150 days old. Fish in the LAZ group spent the same amount of time in the upper zone as the MAZ group at 90 days old, then more time at 120 and 150 days old. D) Similarly, fish in the HAZ group consistently spent more time in the lower zone than the MAZ and LAZ groups at 90, 120, and 150 days old. Fish in the LAZ group spent the same amount of time in the lower zone as the MAZ group at 90 days old, but less time in the lower zone at 120 and 150 days old. All data are presented as mean±S.E.M. Anxiety-level groups were assigned based on lower zone ranking at 150 dpf. Significant differences between groups are indicated by ** (p<0.01), *** (p<0.001), and **** (p<0.0001), as well as different letters above bars (groups not sharing a letter differ significantly, p<0.05).

anxiety-like behaviours spent more time in the upper half of the novel tank during single-trial exposures. The observed increase in swimming velocity at 150 dpf indicates age-related alterations in locomotor activity, which may reflect greater exploratory behaviour or reduced stress. This finding contrasts with earlier studies that reported age-related declines in locomotion [25]. The divergence may be due to differences in testing paradigms or environmental conditions, such as tank dimensions, which have been shown to influence zebrafish activity levels [14,33]. Changes in housing density may also contribute to variation in swimming behaviour through effects on body size, as reduced density typically promotes growth. While increased body size is often associated with greater swimming velocity, studies show this relationship is not linear and may depend on morphology, sex, and physiological factors. Furthermore, both housing density and tank size can modulate swimming behaviour, underscoring the importance of considering environmental context when interpreting loco-motor data [41–43]. In the present study, only males showed increased swimming velocity across time, whereas females' velocities remained stable, suggesting that sex-specific differences and environmental effects combine to influence swimming performance. Furthermore, the reduction in immobility time at 150 dpf supports the interpretation that zebrafish exhibit reduced anxiety and increased locomotor activity as they age. These results are consistent with work by Gerlai et al. [15], who demonstrated that anxiety-reducing interventions, such as ethanol exposure, similarly decrease immobility

and increase swimming velocity in zebrafish. The parallels between pharmacological and age-related effects underscore the importance of controlling for age as a potential confounding factor in drug screening studies.

## 4.2. Sex differences

Our analysis revealed significant sex differences in locomotor behaviours, with males exhibiting higher swimming velocities and greater within-individual variability compared to females. These findings align with previous studies which report that male zebrafish generally display greater locomotor activity levels than females [25,44], possibly reflecting differences in stress coping styles [45]. However, other work has shown that female zebrafish may also exhibit greater activity alongside increased anxiety-like behaviour in the novel tank test [26,46], highlighting the complexity of sex-specific behavioural profiles. Interestingly, while males' swimming velocity increased with age, females exhibited stable locomotor activity across all testing periods. This difference could be related to sex-specific behavioural strategies, as males often demonstrate heightened activity in response to novel or competitive environments [47,48], while females tend to show more consistent patterns across contexts [22]. Although no significant sex differences were observed in anxiety-like measures (e.g., time spent in the lower zone), the observed variability in locomotor behaviours demonstrates the need to account for sex as a key factor in behavioural analyses.

## 4.3. Individual variation

Analyses of individual variation revealed that zebrafish became more behaviourally distinct from one another with age, as among-individual variation increased across anxiety-like and locomotor measures. At the same time, within-individual variation decreased, indicating that individual fish displayed greater consistency in their behaviours over time. This pattern indicates that anxiety-related and locomotor traits become more stable and trait-like as zebrafish mature, supporting previous findings that behavioural repeatability increases with age [29]. These results are consistent with Rajput et al. [44], who reported that exploratory behaviour in zebrafish remains stable over time and is shaped by both strain and sex. Sex differences in individual variation were also evident. Females exhibited higher repeatability in anxiety-like behaviours and locomotor traits, indicating that they were more behaviourally stable across repeated tests, a trend counter to prior studies on zebrafish stress responses [47]. Thomson et al. [29] found that male zebrafish exhibit higher repeatability than females, a result consistent with broader studies in other species [49]. In contrast, males showed greater behavioural plasticity, particularly in locomotor measures, with increasing divergence between individuals over time. While both sexes became more distinct from each other at the group level, males retained higher within-individual variability in locomotion, suggesting greater sensitivity to environmental influences or generally less consistent movement responses. Given the observed variability in anxiety-like behaviour, it is essential to consider these differences when interpreting experimental outcomes, as they may influence responses to pharmacological manipulations and behavioural assays [22]. For example, the effects of ethanol in zebrafish have been shown to be modulated by the coping style of the fish, where fish with a proactive coping style spent significantly more time in the upper zone of the arena following ethanol administration than fish with a reactive coping style [50]. Similarly, Beigloo et al. [51] demonstrated that bold and shy female zebrafish exhibit opposing behavioural responses to an anxiolytic compound, highlighting how intrinsic personality traits can meaningfully modulate pharmacological effects, and how variability in novelty response is influenced by sex and developmental stage. Together, these findings suggest that anxiety phenotype may critically influence drug responsiveness and should be considered in the interpretation of pharmacological studies using zebrafish. The increasing stability of anxiety-like behaviours with age supports their consideration as reliable individual traits, but the concurrent increase in among-individual variation highlights the need to account for natural behavioural diversity when interpreting experimental outcomes. Additionally, sex differences in behavioural repeatability suggest that males and females may respond differently to pharmacological interventions, potentially affecting drug sensitivity and efficacy.

## 4.4. Group variation

Classifying zebrafish into high-, medium-, and low-anxiety groups revealed that behavioural profiles were relatively stable across time. Fish in the high-anxiety (HAZ) group consistently spent more time in the lower zone, while those in the low-anxiety (LAZ) group spent more time in the upper zone. These findings align with previous research that demonstrate individual differences in zebrafish behaviours are maintained across biologically meaningful timescales [44] and testing contexts [31]. Interestingly, the LAZ group exhibited significant increases in upper-zone activity at 120 and 150 dpf compared to 90 dpf, suggesting that baseline anxiety levels may influence behavioural plasticity. This observation could have important implications for pharmacological studies, as individuals with low baseline anxiety may respond differently to anxiolytic compounds, potentially confounding the interpretation of drug effects.

While the anxiety group classifications were applied retrospectively based on behaviour at 150 dpf, the goal was to explore whether distinguishable behavioural profiles had emerged by late adolescence. This approach does not define trait anxiety prospectively but allows for examination of whether consistent group-level patterns were already present earlier in development. Notably, behavioural differences between groups were evident at 90 and 120 dpf, supporting the interpretation that these groupings captured stable behavioural tendencies rather than short-term fluctuations or habituation effects. Given the exploratory nature of this classification, future studies using prospective grouping or selective breeding designs will be important for validating the developmental stability of these phenotypes.

## 5. Conclusion

This study demonstrates that anxiety-like behaviours in zebrafish are shaped by age, sex, and individual variability, with stable, trait-like patterns emerging over time. Classifying individuals into high-, medium-, and low-anxiety groups further supports the existence of consistent behavioural phenotypes and highlights the potential of zebrafish as a model organism for investigating anxiety-related traits. Given the observed stability of individual behaviours, future research should explore selective breeding to establish high- and low-anxiety zebrafish lines, similar to bold and shy phenotypes or stress-coping style [31,52]. These lines could enable targeted studies on the genetic basis of anxiety and provide a more controlled platform for pharmacological testing. Our findings also suggest that baseline anxiety levels may influence drug responses, emphasizing the need to examine how individual differences shape outcomes in anxiolytic drug research. More broadly, these results highlight the value of incorporating individual-level data into experimental designs to improve reproducibility and interpretation in zebrafish-based anxiety models. Accounting for natural behavioural variability may enhance the translational relevance of zebrafish research to human psychiatric conditions and treatment strategies.

## Supporting information

**S1 File. EthoVision data file.**
(XLSX)

## Acknowledgments

We would like to thank Aleah McCorry, the MacEwan Animal Care Coordinator, and MacEwan technicians Kaylee Gehlert, Haley Irwin, and Brittany Miller, for their help with daily husbandry and aquarium maintenance.

## Author contributions

**Conceptualization:** Trevor J. Hamilton.

**Formal analysis:** Andréa L. Johnson, Kimberley J. Mathot, Trevor J. Hamilton.

**Investigation:** Andréa L. Johnson, Trevor J. Hamilton.

**Methodology:** Trevor J. Hamilton.

**Project administration:** Trevor J. Hamilton.

**Resources:** Trevor J. Hamilton.

**Supervision:** Trevor J. Hamilton.

**Visualization:** Andréa L. Johnson, Trevor J. Hamilton.

**Writing – original draft:** Andréa L. Johnson, Kimberley J. Mathot, Trevor J. Hamilton.

**Writing – review & editing:** Andréa L. Johnson, Peter L. Hurd, Kimberley J. Mathot, Trevor J. Hamilton.

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
