## [Decision Letter · Decision Letter 0]

8 Jul 2025

Dear Dr. Hamilton,

Thank you for submitting your manuscript to PLOS ONE. After careful consideration, I invite you to submit a revised version of the manuscript that addresses the points raised during the review process.

Editor comments: Thank you for this interesting manuscript. As an avid user of noldus programs myself, could the lab please clarify the frame rate used for ethovision (and whether this was changed as the fish got older)? 

We look forward to receiving your revised manuscript.

Kind regards,

Benjamin Tsang

Academic Editor

PLOS ONE

Journal Requirements:

“This research received funding from the Natural Sciences and Engineering Research Council of Canada (NSERC) Discovery Development grant to T.J.H. (05426).”

Reviewers' comments:

Reviewer's Responses to Questions

**Comments to the Author**

1. Is the manuscript technically sound, and do the data support the conclusions?

Reviewer #1: Yes

Reviewer #2: Partly

Reviewer #3: Yes

2. Has the statistical analysis been performed appropriately and rigorously?

Reviewer #1: Yes

Reviewer #2: Yes

Reviewer #3: I Don't Know

3. Have the authors made all data underlying the findings in their manuscript fully available?

Reviewer #1: Yes

Reviewer #2: Yes

Reviewer #3: Yes

4. Is the manuscript presented in an intelligible fashion and written in standard English?

Reviewer #1: Yes

Reviewer #2: Yes

Reviewer #3: Yes

Reviewer #1: Johnson and colleagues present a nice study of zebrafish behavior across time. They examine the variability and repetability of zebrafish behavior across three ages (90, 120, and 150 dpf) in the novel tank test. Overall, the manuscript is well written and the results are clear. My comments are mostly minor to improve data presentation, to point out some more recent citations they may want to incorporate, and to clarify some of the statistics.

Minor points

Line 136. I believe luminance is measured as candela per meter squared (not meter cubed) as it refers to a surface.

Lines 174 and 180. These methods refer to 'period 1' and 'period 2'. However, in the results section and figure (figure 5) the terms 'early' and 'late' are used. I would keep these consistent throughout the paper to make it easier for the reader to understand what 'early' and 'late' refer to when reading the results

Lines 189-190 and figure 5. The author's state that significant differences between groups are when the 95% CrIs do not overlap. However, in figure 5 it appears that several 95% CrIs overlap yet the author's state that there are differences between groups. For example, in figure 5A, the results (lines 257-259) state that there is a difference between males and females. But the 95% CrI's overlap. So this does not appear to be a significant difference. However, I'm not that familiar with Bayesian statistics, and so perhaps I'm misunderstanding something here. If so, I would suggest the author's be a bit clearer about what constitutes a 'significant' or relevant difference in the results section.

Line 331. The author's state that "all n=100 fish were then ranked by time spent...". However, I got the impression earlier in this section (lines 311-314) that only 30 fish were used (10 in each group of HAZ, LAZ, and MAZ). Or was it that only the 'top 10, bottom 10, and middle-most 10' fish were used for the different groups. If so, that wasn't entirely clear from the writing in this section.

It is not clear when different post-hoc tests are used in different figures/results. In the methods section its stated that sometimes a Dunnet (line 156) or Tukey HSD (line 157) or Fisher's LSD (line 165) is used as a post-hoc test. However, it's not clear in the results when the different tests are used. This is important because, for example, the Fisher's LSD does not manage family wise error rates like the Dunnett or Tukey does. This will help readers better interpret which effects are likely to be more robust.

Discussion

Line 353-355. The increase in velocity could also be due to an increase in size. Especially if fish are being housed at the low density indicated here, they are likely to grow quite a bit more between 90 and 120 dpf if they were housed at higher density beforehand. This latter point should also be made.

Finally, I would also kindly point the author's to papers we have recently published that touches on several of the topics discussed in this paper. For example, in Rajput et al, 2022 we explicitly study the consistency of individual differences over time and look at strain and sex differences. We show, for example, that males have higher locomotor activity than females (line 368) and that individual differences in behavior are largely consistent over days and weeks (lines 80, 85, and 92). In another paper (Beigloo et al, 2024), we also show that behavioral type can affect behavioral responses to anxiolytic compounds (escitalopram), an idea suggested in the discussion (lines 414-417). That being said, because these papers are from my own lab, it is completely at the discretion of the author and editor whether it is appropriate to include these citations.

References

Rajput, N., Parikh, K., & Kenney, J. W. (2022). Beyond bold versus shy: Zebrafish exploratory behavior falls into several behavioral clusters and is influenced by strain and sex. Biology Open, 11(8), bio059443.

Beigloo, F., Davidson, C. J., Gjonaj, J., Perrine, S. A., & Kenney, J. W. (2024). Individual differences in the boldness of female zebrafish are associated with alterations in serotonin function. Journal of Experimental Biology, 227(12).

Reviewer #2: This manuscript presents a longitudinal behavioral analysis of adult zebrafish using the Novel Tank Dive Test (NTT), highlighting individual variability and repeatability across development. The study is methodologically ambitious and holds promise to inform future work on zebrafish anxiety-like behavior. However, there are several major concerns regarding the experimental design, interpretation of results, and reproducibility that must be addressed before publication.

Point 1. Habituation and Repeated Testing Effects is a major concern. Although the authors acknowledge the possibility that behavioral changes over time may result from re-exposure rather than developmental progression, they nonetheless interpret their findings as age-related decreases in anxiety-like behavior. This is methodologically problematic.

Repeated testing in the NTT is well known to produce habituation effects, which can mimic anxiolysis. Therefore, it is inappropriate to infer developmental reductions in anxiety without a control group of naïve animals tested only once at each time point. I strongly recommend the authors to either include additional experiments with naïve age-matched groups or substantially reframe their claims.

Point 2. Lack of Arena Randomization and Positional Control is also a problem. There is no description of how the arenas were assigned or randomized across testing days or fish. Since four fish were tested simultaneously, arena position or time-of-day effects could bias results or artificially inflate repeatability. Please, clarify whether randomization procedures were implemented.

Point 3. The manuscript fails to describe how sex was confirmed. This is especially important in adult zebrafish as external sexual dimorphism can be subtle and unreliable, and gonadal dissection or genetic confirmation is often necessary for accurate assignment. Please clarify how sex was identified, whether through dissection, visual dimorphism, or another method. If not confirmed post-mortem, this introduces potential misclassification bias in sex-based analyses.

Point 4. It is stated that 7 females and 9 males were lost during the study, yet no details are provided regarding the causes, timing, or potential impact on longitudinal behavioral profiles. Given the experimental design, if fish were sharing a tank 1:1, how did the authors proceed with the fish that shared the tank with the one that died? Could that affect the behavior of the remaining fish?

Point 5. The division of fish into “low,” “medium,” and “high” anxiety groups based on 150 dpf data is methodologically problematic. Using endpoint data for retrospective grouping can lead to circular reasoning and ignores behavioral plasticity or adaptation over time. For instance, individuals classified as "low anxiety" may simply be more habituated or more adaptable, not inherently less anxious.

Point 6. Two recent studies that directly relate to the present work are not cited but are essential to contextualize the findings. One study (doi: 10.1242/bio.059443) examined individual exploratory behavior in a novel environment over time, revealing consistent inter-individual differences and reinforcing the importance of accounting for stable behavioral phenotypes in repeated-measures designs. Another study (doi: 10.1371/journal.pone.0300227) found that individual variability in novelty response is influenced by strain, sex, and age, which directly informs the present study’s focus on behavioral repeatability and developmental trajectories. Please revise the reference list to include relevant studies and ensure that other foundational papers on sex differences in zebrafish behavior, particularly in the Novel Tank Test, are cited, as some seminal articles appear to be missing.

Point 7. Anxiety-like behavior in zebrafish, particularly in older individuals, may not follow the same trajectory or behavioral signatures as in larvae or juveniles. The Novel Tank Test (NTT) was originally validated for acute anxiogenic responses, and its use as a developmental or trait-based marker of anxiety in adult zebrafish is still under debate. In this context, it is unclear why the authors chose time spent in tank zones (e.g., upper vs. lower) as the primary behavioral feature to stratify individuals into anxiety-level groups, while other relevant parameters such as immobility, freezing-like behavior, or overall activity were not used as equally weighted indicators. Furthermore, while zone-based scoring is conventional, the use of continuous variables such as distance from bottom may offer a more robust and fine-grained analysis of vertical exploration, without the need to define arbitrary cutoffs or classify subjects post hoc. This would also allow for correlational or trajectory-based analyses of anxiety-like behavior over time, strengthening the claim of trait-level consistency. The authors should justify their prioritization of zone occupancy as the main behavioral classifier, clarify whether immobility or other anxiety-related metrics were considered in combination, and consider re-analyzing or presenting vertical position as a continuous measure rather than a categorical one. This may offer greater sensitivity to developmental or individual differences and avoid potential oversimplification of complex behavioral traits.

Point 8. The manuscript does not provide a rationale for the chosen sample size (n = 100; 50 males and 50 females). While this is a commendably large N for a behavioral longitudinal study, no information is provided regarding power analysis, effect size expectations, or justification based on previous studies.

Reviewer #3: This study assessed individuality and variability across adult zebrafish, with observation spanning across 2 months, using the novel tank dive test. It highlights the need to consider age, sex, and individual variation when conducting zebrafish behavioral assays. Overall, I am very enthusiastic about this manuscript.

Here are my comments:

Were experimental fish housed individually?

Just out of curiosity – how did the authors deal with the egg-bound issue of female fish over this 2-month period?

9 fish were dead, and the data were not included. Thus, the “n” in the results should be reflected accordingly, instead of “n = 50” or “n = 100”.

Could the author comment on why the locomotion was increased at 90 dpf, compared to 120 dpf, given that the fish were younger and presumably smaller at 90 dpf?

I am not sure if I understand the interpretation of Fig.4. The error bars are mostly overlapping. Are these reported trends (such as “decline”, “reduction”) significant? If not, how are the conclusions justified?

Fig 5, if variance was measured here, should the unit of the y-axis be “squared”?

A couple of comments on statistical analysis:

Line 189-191: I am not sure why would non-overlapping 95% credible interval would correspond to p<0.006 (credible interval is based on Bayesian, p value is based on frequentist). Please elaborate.

Could the authors please add more details about what the sim function of the arm package and the MCMCglmm package are and why they are used?

**Do you want your identity to be public for this peer review?** For information about this choice, including consent withdrawal, please see our Privacy Policy

Reviewer #1: **Yes: ** Justin Kenney

Reviewer #2: No

Reviewer #3: No

---

## [Author Response · Author response to Decision Letter 1]

20 Sep 2025

Dear Academic Editor and Expert Reviewers,

Thank you for the time and effort you have put into suggesting edits and changes to our manuscript. We have addressed all comments below in bold on the document attached. We believe the manuscript is much improved.

Sincerely,

Johnson and colleagues

PONE-D-25-29898

Behavioural Variability and Repeatability in Adult Zebrafish (Danio rerio) Using the Novel Tank Dive Test

PLOS ONE

Editor comments: Thank you for this interesting manuscript. As an avid user of noldus programs myself, could the lab please clarify the frame rate used for ethovision (and whether this was changed as the fish got older)?

We used a frame rate of 25 frames per second throughout the experiment and have added this to the methods section.

Journal Requirements:

Revised.

“This research received funding from the Natural Sciences and Engineering Research Council of Canada (NSERC) Discovery Development grant to T.J.H. (05426).”

The suggested statement has been added to the financial disclosure.

5. Review Comments to the Author

Reviewer #1: Johnson and colleagues present a nice study of zebrafish behavior across time. They examine the variability and repetability of zebrafish behavior across three ages (90, 120, and 150 dpf) in the novel tank test. Overall, the manuscript is well written and the results are clear. My comments are mostly minor to improve data presentation, to point out some more recent citations they may want to incorporate, and to clarify some of the statistics.

Minor points

Line 136. I believe luminance is measured as candela per meter squared (not meter cubed) as it refers to a surface.

Thank you for spotting this typo. It has been corrected (line 163).

Lines 174 and 180. These methods refer to 'period 1' and 'period 2'. However, in the results section and figure (figure 5) the terms 'early' and 'late' are used. I would keep these consistent throughout the paper to make it easier for the reader to understand what 'early' and 'late' refer to when reading the results

Thank you for catching this oversight. This has been amended to refer to period 1 and 2 as early and late (line 202).

Lines 189-190 and figure 5. The author's state that significant differences between groups are when the 95% CrIs do not overlap. However, in figure 5 it appears that several 95% CrIs overlap yet the author's state that there are differences between groups. For example, in figure 5A, the results (lines 257-259) state that there is a difference between males and females. But the 95% CrI's overlap. So this does not appear to be a significant difference. However, I'm not that familiar with Bayesian statistics, and so perhaps I'm misunderstanding something here. If so, I would suggest the author's be a bit clearer about what constitutes a 'significant' or relevant difference in the results section.

Thank you for this question. The proportion overlap between the 95% CrI of two independent estimates provides a conservative estimate of p-value [39], and thus, effects whose pr < 0.05 are also described as being significantly different. We have added this detail to the methods section (line 227-230).

Line 331. The author's state that "all n=100 fish were then ranked by time spent...". However, I got the impression earlier in this section (lines 311-314) that only 30 fish were used (10 in each group of HAZ, LAZ, and MAZ). Or was it that only the 'top 10, bottom 10, and middle-most 10' fish were used for the different groups. If so, that wasn't entirely clear from the writing in this section.

Thank you, this was not worded very well. You are correct. It has been revised to explain that 30 fish were selected from the sample (of N = 91) to form the HAZ, MAZ, and LAZ groups (top 10, bottom 10, and middle-most 10), then their behaviour was analyzed to identify any distinct patterns between groups. “To determine whether distinct high, medium, and low anxiety-level groups could be established based on their zone preference, we performed post-hoc rankings based on the cumulative time each fish (N = 91) spent in the bottom zone of the arena at 150 dpf. The 10 male and female zebrafish who spent the most time in the bottom zone were labelled high anxiety (HAZ), while the 10 zebrafish who spent the least time in the bottom zone were labelled low anxiety (LAZ) and the 10 zebrafish falling in the middle were labelled medium anxiety fish (MAZ)” (line 361-367).

Afterwards the entire N = 91 sample was divided into HAZ, MAZ, and LAZ groups to see if similar patterns of behaviour were present. However, this was not necessary and seemed to cause confusion, so we have decided to remove this section and associated figures.

It is not clear when different post-hoc tests are used in different figures/results. In the methods section its stated that sometimes a Dunnet (line 156) or Tukey HSD (line 157) or Fisher's LSD (line 165) is used as a post-hoc test. However, it's not clear in the results when the different tests are used. This is important because, for example, the Fisher's LSD does not manage family wise error rates like the Dunnett or Tukey does. This will help readers better interpret which effects are likely to be more robust.

The statistical analysis section has been revised to better clarify which analyses were done for each results section.

Line 180-196: “For each behaviour variable, data distributions were assessed using the D’Agostino-Pearson omnibus normality test. Because repeated measures across time within subjects violated normality assumptions, age-related differences were evaluated using the non-parametric Friedman test, followed by Dunn’s multiple comparisons test to identify significant pairwise differences across timepoints. To assess sex differences and age × sex interactions, a two-way ANOVA was conducted for each behavioural measure, treating age and sex as between-subjects factors. Post hoc comparisons were performed using Fisher’s Least Significant Difference (LSD) test. These analyses did not account for within-subject matching and were treated as independent group comparisons across age and sex. Although the data were non-normally distributed, ANOVAs were applied due to their robustness to moderate deviations from normality in balanced designs with large sample sizes. To assess potential effects of repeated exposure, naïve and longitudinal groups were compared at each developmental stage (90, 120, 150 dpf). Because Bartlett’s test indicated unequal variances, Brown–Forsythe ANOVAs were used for these comparisons, followed by Dunnett’s T3 post hoc test. An alpha level of p < .05 and 95% confidence intervals were used to assess statistical significance. All values are presented as mean ± standard error of the mean (S.E.M.).”

Discussion

Line 353-355. The increase in velocity could also be due to an increase in size. Especially if fish are being housed at the low density indicated here, they are likely to grow quite a bit more between 90 and 120 dpf if they were housed at higher density beforehand. This latter point should also be made.

Thank you for highlighting how changes in housing density may influence swimming velocity via body size, and for prompting consideration of appropriate references. Recent studies are now cited to support this, and environmental context is emphasized as important in interpreting velocity data. To address this, the Discussion has been amended to read; “Changes in housing density may also contribute to variation in swimming behaviour through effects on body size, as reduced density typically promotes growth. While increased body size is often associated with greater swimming velocity, studies show this relationship is not linear and may depend on morphology, sex, and physiological factors. Furthermore, both housing density and tank size can modulate swimming behaviour, underscoring the importance of considering environmental context when interpreting locomotor data [41-43]. In the present study, only males showed increased swimming velocity across time, whereas females’ velocities remained stable, suggesting that sex-specific differences and environmental effects combine to influence swimming performance” (lines 421-430).

Shishis, S., Tsang, B., & Gerlai, R. (2022). The effect of fish density and tank size on the behavior of adult zebrafish: A systematic analysis. Frontiers in Behavioral Neuroscience, 16. https://doi.org/10.3389/fnbeh.2022.934809

Palstra, A. P., Tudorache, C., Rovira, M., Brittijn, S. A., Burgerhout, E., van den Thillart, G. E., Spaink, H. P., & Planas, J. V. (2010). Establishing zebrafish as a novel exercise model: Swimming Economy, swimming-enhanced growth and muscle growth marker gene expression. PLoS ONE, 5(12). https://doi.org/10.1371/journal.pone.0014483

Miller, C. L., Dugand, R., & McGuigan, K. (2024). Variability of morphology–performance relationships under acute exposure to different temperatures in 3 strains of zebrafish. Current Zoology, 71(2), 152–161. https://doi.org/10.1093/cz/zoae032

Finally, I would also kindly point the author's to papers we have recently published that touches on several of the topics discussed in this paper. For example, in Rajput et al, 2022 we explicitly study the consistency of individual differences over time and look at strain and sex differences. We show, for example, that males have higher locomotor activity than females (line 368) and that individual differences in behavior are largely consistent over days and weeks (lines 80, 85, and 92). In another paper (Beigloo et al, 2024), we also show that behavioral type can affect behavioral responses to anxiolytic compounds (escitalopram), an idea suggested in the discussion (lines 414-417). That being said, because these papers are from my own lab, it is completely at the discretion of the author and editor whether it is appropriate to include these citations.

References

Rajput, N., Parikh, K., & Kenney, J. W. (2022). Beyond bold versus shy: Zebrafish exploratory behavior falls into several behavioral clusters and is influenced by strain and sex. Biology Open, 11(8), bio059443.

Beigloo, F., Davidson, C. J., Gjonaj, J., Perrine, S. A., & Kenney, J. W. (2024). Individual differences in the boldness of female zebrafish are associated with alterations in serotonin function. Journal of Experimental Biology, 227(12).

Thank you for directing us to your recent publications. We read both Rajput et al. (2022) and Beigloo et al. (2024) with interest and found them highly relevant to our discussion of individual behavioural consistency, sex differences, and the influence of behavioural phenotype on pharmacological responsiveness. We felt these studies contributed to our interpretation of the results, particularly with regard to trait stability across time and the potential modulatory role of personality traits in drug effects. We have now cited both papers in the revised manuscript and integrated their findings into the discussion.

On line 439, we added the following: “These findings align with previous studies which report that male zebrafish generally display greater locomotor activity levels than females [25, 44], possibly reflecting differences in stress coping styles [56].

On line 456: “This pattern indicates that anxiety-related and locomotor traits become more stable and trait-like as zebrafish mature, supporting previous findings that behavioural repeatability increases with age [29]. These results are consistent with Rajput et al. [44], who reported that exploratory behaviour in zebrafish remains stable over time and is shaped by both strain and sex.”

On line 475: “Similarly, Beigloo et al. [51] demonstrated that bold and shy female zebrafish exhibit opposing behavioural responses to an anxiolytic compound, highlighting how intrinsic personality traits can meaningfully modulate pharmacological effects, and how variability in novelty response is influenced by sex and developmental stage. Together, these findings suggest that anxiety phenotype may critically influence drug responsiveness and should be considered in the interpretation of pharmacological studies using zebrafish.”

On line 490: “These findings align with previous research that demonstrate individual differences in zebrafish behaviours are maintained across biologically meaningful timescales [44] and testing contexts [31].”

Reviewer #2: This manuscript presents a longitudinal behavioral analysis of adult zebrafish using the Novel Tank Dive Test (NTT), highlighting individual variability and repeatability across development. The study is methodologically ambitious and holds promise to inform future work on zebrafish anxiety-like behavior. However, there are several major concerns regarding the experimental design, interpretation of results, and reproducibility that must be addressed before publication.

Point 1. Habituation and Repeated Testing Effects is a major concern. Although the authors acknowledge the possibility that behavioral changes over time may result from re-exposure rather than developmental progression, they nonetheless interpret their findings as age-related decreases in anxiety-like behavior. This is methodologically problematic.

Repeated testing in the NTT is well known to produce habituation effects, which can mimic anxiolysis. Therefore, it is inappropriate to infer developmental reductions in anxiety without a control group of naïve animals tested only once at each time point. I strongly recommend the authors to either include additional experiments with naïve age-matched groups or substantially reframe their claims.

We thank the reviewer for raising the important issue of habituation and repeated testing effects in the Novel Tank Dive Test (NTT). We fully acknowledge that in some tests with short intervals between data collection, repeated exposures can induce habituation, potentially mimicking anxiolytic effects. To address this concern and further support our interpretation, we note that our testing intervals were substantial (approximately one month between sessions) rather than massed or frequent exposures. The available literature on larval zebrafish demonstrates that the persistence of habituation is strongly dependent on the interval between exposures: studies have shown that habituation to aversive or novelty-based stimuli in zebrafish larvae decays or even reverses with long intertrial interv

---

## [Decision Letter · Decision Letter 1]

10 Oct 2025

Behavioural Variability and Repeatability in Adult Zebrafish (Danio rerio) Using the Novel Tank Dive Test

PONE-D-25-29898R1

Dear Dr. Hamilton,

We’re pleased to inform you that your manuscript has been judged scientifically suitable for publication and will be formally accepted for publication once it meets all outstanding technical requirements.

Kind regards,

Benjamin Tsang

Academic Editor

PLOS ONE

Additional Editor Comments (optional):

Reviewers' comments:

Reviewer's Responses to Questions

**Comments to the Author**

Reviewer #1: All comments have been addressed

Reviewer #2: All comments have been addressed

Reviewer #3: All comments have been addressed

2. Is the manuscript technically sound, and do the data support the conclusions?

Reviewer #1: Yes

Reviewer #2: Yes

Reviewer #3: Yes

3. Has the statistical analysis been performed appropriately and rigorously?

Reviewer #1: Yes

Reviewer #2: Yes

Reviewer #3: Yes

4. Have the authors made all data underlying the findings in their manuscript fully available?

Reviewer #1: Yes

Reviewer #2: Yes

Reviewer #3: Yes

5. Is the manuscript presented in an intelligible fashion and written in standard English?

Reviewer #1: Yes

Reviewer #2: Yes

Reviewer #3: Yes

Reviewer #1: All of my concerns were adequately addressed.

Reviewer #2: The authors have responded thoroughly to all my concerns, including the addition of new analyses and data (e.g., naïve cohort, power analysis, continuous metrics). I now recommend the manuscript for publication in its current form.

Reviewer #3: (No Response)

**Do you want your identity to be public for this peer review?** For information about this choice, including consent withdrawal, please see our Privacy Policy

Reviewer #1: No

Reviewer #2: No

Reviewer #3: **Yes: ** Qian Lin

---

## [Editor Report · Acceptance letter]

PONE-D-25-29898R1

PLOS ONE

Dear Dr. Hamilton,

I'm pleased to inform you that your manuscript has been deemed suitable for publication in PLOS ONE. Congratulations! Your manuscript is now being handed over to our production team.

Kind regards,

on behalf of

Dr. Benjamin Tsang

Academic Editor

PLOS ONE